# Kapitza-resistance-like exciton dynamics in atomically flat MoSe₂-WSe₂ lateral heterojunction

Hassan Lamsaadi [1], Dorian Beret[2], Ioannis Paradisanos [2,3], Pierre Renucci[2], Delphine Lagarde[2], Xavier Marie [2], Bernhard Urbaszek[2,4], Ziyang Gan[5], Antony George [5,6], Kenji Watanabe [7], Takashi Taniguchi [8], Andrey Turchanin [5,6], Laurent Lombez [2] ✉, Nicolas Combe [1], Vincent Paillard[1] & Jean-Marie Poumirol [1] ✉

Being able to control the neutral excitonic flux is a mandatory step for the development of future room-temperature two-dimensional excitonic devices. Semiconducting Monolayer Transition Metal Dichalcogenides (TMD-ML) with extremely robust and mobile excitons are highly attractive in this regard. However, generating an efficient and controlled exciton transport over long distances is a very challenging task. Here we demonstrate that an atomically sharp TMD-ML lateral heterostructure (MoSe₂-WSe₂) transforms the isotropic exciton diffusion into a unidirectional excitonic flow through the junction. Using tip-enhanced photoluminescence spectroscopy (TEPL) and a modified exciton transfer model, we show a discontinuity of the exciton density distribution on each side of the interface. We introduce the concept of exciton Kapitza resistance, by analogy with the interfacial thermal resistance referred to as Kapitza resistance. By comparing different heterostructures with or without top hexagonal boron nitride (hBN) layer, we deduce that the transport properties can be controlled, over distances far greater than the junction width, by the exciton density through near-field engineering and/or laser power density. This work provides a new approach for controlling the neutral exciton flow, which is key toward the conception of excitonic devices.

Electronics relies on the control of the motion of charge carriers to process information. The losses caused by the charged particle current and the resulting need to improve the power efficiency have fueled lots of interest in recent years[1,2]. As it is based on the control of electrically neutral quasi-particles (excitons), insensitive to long-range Coulomb scattering mechanisms, excitronics is by nature much more power efficient as it presents only negligible ohmic losses[3,4]. Nevertheless, developing excitronic devices is challenging, as it requires the ability to control the neutral exciton properties, such as the recombination rates, diffusion length ($L_D$) and propagation direction, in an optically active medium at room temperature, without the help of external electric or magnetic fields[3,5].

[1]CEMES-CNRS, Université de Toulouse, Toulouse, France. [2]Université de Toulouse, INSA-CNRS-UPS, LPCNO, 135 Avenue Rangueil, 31077 Toulouse, France. [3]Institute of Electronic Structure and Laser, Foundation for Research and Technology-Hellas, Heraklion 70013, Greece. [4]Institute of Condensed Matter Physics, Technische Universität Darmstadt, Darmstadt, Germany. [5]Friedrich Schiller University Jena, Institute of Physical Chemistry, 07743 Jena, Germany. [6]Abbe Centre of Photonics, 07745 Jena, Germany. [7]Research Center for Functional Materials, National Institute for Materials Science, 1-1 Namiki, Tsukuba 305-0044, Japan. [8]International Center for Materials Nanoarchitectonics, National Institute for Materials Science, 1-1 Namiki, Tsukuba 305-0044, Japan. ✉e-mail: laurent.lombez@insa-toulouse.fr; jean-marie.poumirol@cemes.fr

Owing to their promising optical properties, Transition Metal Dichalcogenide monolayers emerged as a highly versatile platform for excitonics system at the nanoscale[6–11]. In particular, their large exciton binding energy allows operating at room temperature. Due to their unique band properties, excitonic transport in TMD-MLs has led to the discovery of new fundamental phenomena such as a valley hall effect[12–14], or the observation of nonlinear behavior such as a halo in the spatial profile[15] or negative effective diffusion[16]. With the increasing maturity of the field, several basic components necessary to control exciton information have been developed, like room temperature excitonic transistor in a Van der Waals vertical heterostructure[17], or an excitonic diode able to filter excitons in lateral heterostructures[18–20]. The physical mechanisms involved in the exciton transfer processes through lateral heterojunctions, as well as their influence on the exciton distribution, dynamics and the resulting photoluminescence observed in previous work[18] are still unclear. This is mainly due to: (i) the extreme sharpness of lateral heterojunctions making complex its optical characterization, (ii) the strong influence of external parameters like defects, strain and dielectric environment on exciton transport, making the fabrication of high-quality samples with controlled dielectric environment crucial to extract the intrinsic junction properties, (iii) the lack of a complete theoretical description able to describe the asymmetric exciton transfer through the junction. Exciton diffusion at sharp interfaces have been previously studied in other materials like organic semiconductors in so-called bulk-heterojunction[21,22]. In the context of solar cells, efficient fast exciton diffusion and harvesting have stimulated strong theoretical and experimental efforts. Nevertheless, in such system the interface is designed to dissociate the exciton into free charges and not to transfer the entire quasi-particle through the junction[21,22]. Up to now, in TMD-ML the exciton diffusion is mainly driven either by strain[23,24] or dielectric gradient engineering in TMD based vertical heterostructure[19]. Both approaches require complex architectures, with nanometric precision of the strain or the electrostatic potential over large distances (micrometers). The fabrication of those excitonic guides as well as their coupling with other circuit elements is thus very challenging.

In this paper, we investigate experimentally and theoretically the effect of an atomically sharp $MoSe_2$-$WSe_2$ lateral heterostructure (LH) on exciton diffusion and distribution. We performed tip-enhanced photoluminescence (TEPL) spectroscopy experiments, allowing sub-wavelength spatial resolution down to 30 nm, and developed an exciton transfer model. We show that the difference in the energy gap at the LH generates a discontinuity in the exciton density distribution analog to the temperature discontinuity found at interfaces presenting thermal resistance (Kapitza resistance). In steady state conditions, the presence of this discontinuity results in unique non-reciprocal exciton transport properties, taking place over distances far greater (two orders of magnitude) than the junction width and experimentally evidenced by: a highly asymmetric photoluminescence (PL) profile, the quenching of the $WSe_2$-related PL and an enhancement of the $MoSe_2$-related PL. Furthermore, by comparing the diffusion properties of fully hBN-encapsulated LH and hBN supported LH (without top hBN layer), we demonstrate that the diffusion properties of the LH can be tuned, by the generated exciton population either by increasing the laser power density or by modifying the optical near-field configuration (at constant laser power).

## Experimental results
### Sample preparation and characterization
The high quality monolayer $MoSe_2$-$WSe_2$ LH is grown using a modified CVD method described in Ref. [25]. We then use water-assisted deterministic transfer to pick up the LH from the as-grown substrate and transfer it on a supporting flake of exfoliated hBN on $SiO_2$/Si substrate. Finally, a second exfoliated hBN flake is transferred to cover the

structure partially[26]. As a result, we obtain two distinct areas, as shown in Fig. 1a, a fully encapsulated area hBN/$WSe_2$-$MoSe_2$/hBN/$SiO_2$/Si (e-LH), and an uncapped area $WSe_2$-$MoSe_2$/hBN/$SiO_2$/Si (un-LH). The dashed white and yellow line in Fig. 1a highlight the boundaries of the bottom and top hBN flakes, respectively.

Figure 1c, f displays the room temperature $\mu$-PL spectra measured on $WSe_2$ and $MoSe_2$, far from the junction on the encapsulated (e-LH) and uncapped (un-LH) regions, respectively. For the e-LH zone, the PL spectrum measured on $MoSe_2$ ML exhibits a pronounced peak centered around 1.575 eV with a full width at half maximum (FWHM) of 50 meV that, in agreement with previous room temperature measurements[27], can be assigned to the neutral exciton ($A_{1s}^{MoSe_2}$). In the case of $WSe_2$, the PL spectrum is asymmetric, due to an intense peak linked to the $A_{1s}^{WSe_2}$ transition located around 1.65 eV (FWHM ~ 30 meV) and a second weaker peak ~ 40 meV below $A_{1s}$, attributed to the spin-forbidden dark exciton ($X_D$)[27]. For the un-LH region, all previously described PL features appear. Furthermore, we observe a slight increase of the broadening of all transitions, of the order of 5%. One can also notice that all PL spectra are more asymmetric. This could be attributed to an increased contribution of the charged excitons (trions), as the unprotected sample can get chemically charged when exposed to ambient air. In any case, one can notice that the integrated PL intensity $I_{PL}^{\infty}(A_{1s}^{WSe_2})$ measured far from the interface is ≈3 times stronger than $I_{PL}^{\infty}(A_{1s}^{MoSe_2})$. Figure 1d, g is spectrally integrated $\mu$-PL intensity color maps. The red (respectively blue) color intensity is the value of integrated PL (shaded areas in c and f) from 1.620 eV to 1.650 eV (respectively from 1.550 eV to 1.580 eV). Figure 1d (g) corresponds to the mapping of the encapsulated (uncapped) region. One can clearly see that the $MoSe_2$ layer is organized in 6 folded stars surrounded by $WSe_2$. The white regions, signature of low emission regions, have different origin. They correspond to bilayer inclusions when located at the center of $MoSe_2$ stars, but also reveal cracks or/and inclusions that are related to the transfer process. We used the far-field PL color maps in combination with Raman mappings to select interfaces away from any visible defects.

### Near-field studies of the lateral heterojunction
In this type of LH, the junction between the two materials $WSe_2$ and $MoSe_2$ is extremely sharp, down to ~3 nm as measured by electron microscopy[18]. Therefore, in order to gain insight into the transport properties around the junction, we use a sub-wavelength resolution tool, TEPL imaging and spectroscopy. Figure 2a displays a schematic of the experimental set-up, showing the linearly polarized laser excitation of 633 nm wavelength (≈1.96eV) focused onto the apex of an atomic force microscope (AFM) silver-coated tip. The exciton generation profile is then linked to the electric field exaltation under the tip, a Gaussian profile with a full width at half maximum of ~30 nm, and a tip position ($x_{tip}$) controlled with AFM resolution. As illustrated in the Fig. 2a excitons diffuse away from the excitation spot over long distances before recombining. The collection of emitted photons is ensured by a long working distance high numerical aperture (0.7 NA) microscope objective, with a fully open collection aperture. The position of the junction is determined very precisely (~30 nm) using tip-enhanced Raman spectroscopy (TERS), as described in ref. 18 and supplementary information (Fig. 6).

TEPL line scans with 30 nm step size are obtained by scanning the tip along a line perpendicularly to the LH junction. Figure 2b, d displays typical TEPL spectra measured for three tip positions ($x_{tip}$) along the measured line for both encapsulated and uncapped samples. The spectra (1) and (3) recorded far from the interface (500 nm inside the $MoSe_2$ region and 1.25 $\mu$m inside the $WSe_2$ region) are similar to the ones described in Fig. 1. The spectra of $WSe_2$ in locations (2) are recorded at 100 nm (respectively 300 nm) from the interface for

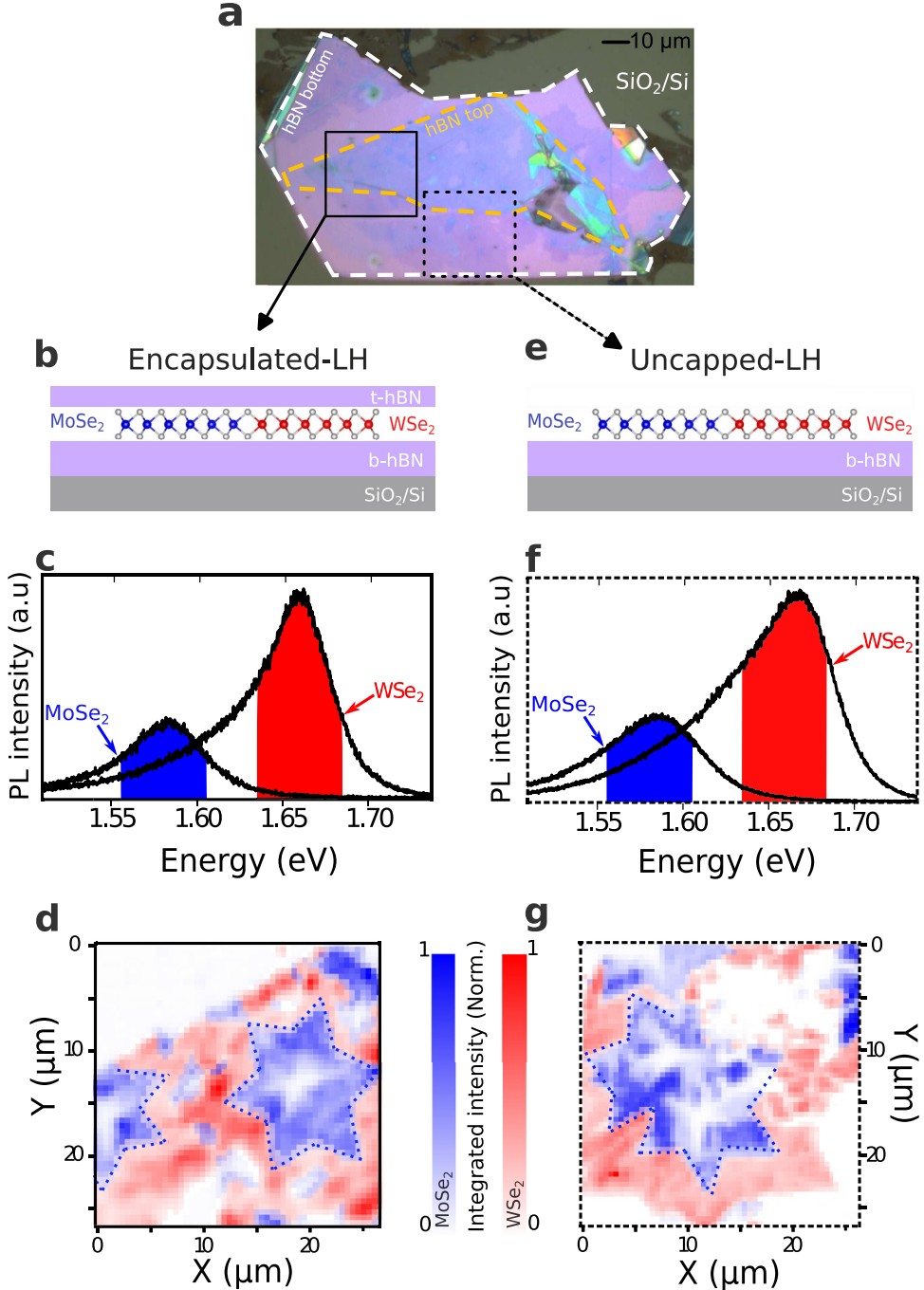

**Fig. 1 | Lateral heterostructure far field optical characterization. a** Optical image of the sample, the white (yellow) dashed contour shows the bottom (top) hBN flake boundaries. The continuous (respectively dashed) black square contour highlights the MoSe₂-WSe₂ LH, encapsulated in hBN (e-LH) region (respectively the hBN supported LH (un-LH) region). **b (e)** Schematic representation of the e-LH (un-LH). **c (f)** typical $\mu$-PL spectra measured in the WSe₂ and MoSe₂ regions of e-LH (un-LH).

**d (g)** Spectrally integrated PL intensity maps of e-LH (un-LH). The PL intensity is obtained by integrating MoSe₂ (WSe₂) PL spectra over the spectra range represented by the blue (red) shaded area. PL spectra are recorded every 500 nm (step size), using a 633 nm ($\approx$ 1.96eV) excitation laser, 400 $\mu$W laser power. The dotted blue lines in (**d**) and (**g**) highlight the boundaries between the two materials.

the e-LH (resp. un-LH) sample. They show contributions of both MoSe₂ and WSe₂.

All TEPL spectra can be fitted using three Lorentzian peaks to account for the previously described relative contributions of $A_{1s}^{WSe_2}$ (dashed red line), $X_{WSe_2}^D$ (dashed green line) and $A_{1s}^{MoSe_2}$ (dashed blue line) as illustrated by the diagram in Fig. 2a. Figure 2c, e displays the result of this fitting procedure. The peak positions and FWHMs (shaded area) of the bright excitons $A_{1s}^{MoSe_2}$ and $A_{1s}^{WSe_2}$ are shown as a function of the tip position $x_{tip}$ in the upper panels, while the

integrated PL intensity ($I_{PL}$) appears in the lower panels. For clarity, the contribution of $X_{WSe_2}^D$ is not displayed (see supplementary information). First, we point out that the $A_{1s}^{MoSe_2}$ signature (blue star) is clearly observed while the excitation is taking place inside WSe₂, ($x_{tip} > 0$, right side of the solid gray line in Fig. 2c, e). A contrario, no signature of $A_{1s}^{WSe_2}$ is observed when the excitation takes place in MoSe₂ ($x_{tip} < 0$, left side of the solid gray line). This reveals that nonreciprocal filtering is taking place at the interface, allowing the excitons to cross the junction from WSe₂ to MoSe₂. The other

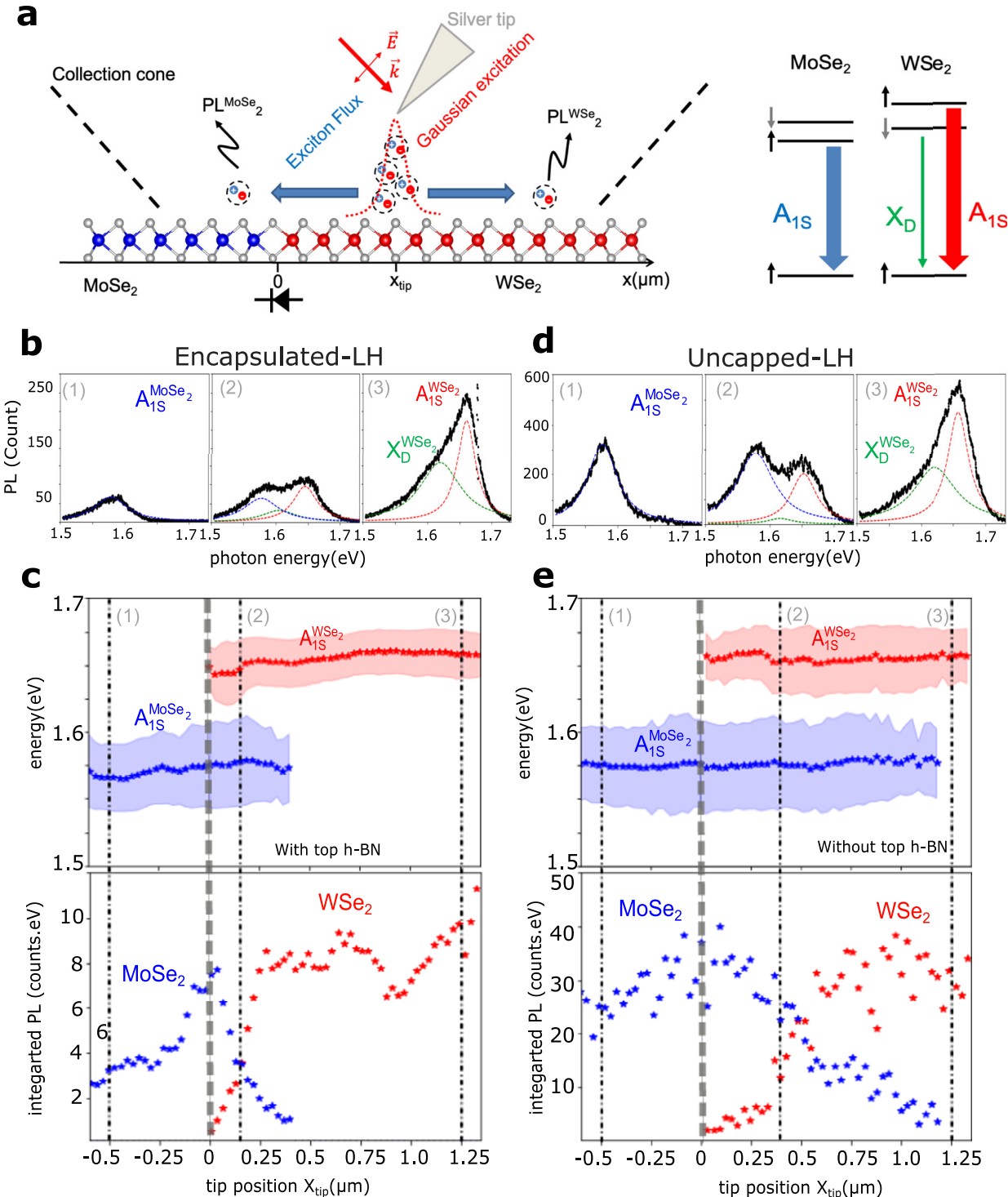

**Fig. 2 | Tip-enhanced spectroscopy of the MoSe₂-WSe₂ interface. a** Left panel: Schematic of the lateral heterojonction, TEPL measurement and the resulting excitonic diffusion properties. Right panel: Diagram of the different excitonic processes observed in our system. **b** (**d**) Typical TEPL spectra taken across the interface in e-LH (respectively un-LH) (1) 500 nm to the left of the interface, (2) 100 nm (300 nm) and (3) 1,25 $\mu$m to the right of the interface. The excitonic contributions are fitted using individual Lorentzian function, neutral WSe₂ exciton in red ($A_{1s}^{WSe_2}$), neutral MoSe₂ exciton in blue ($A_{1s}^{MoSe_2}$) and the dark exciton (out-of-plan) in green ($X_D^{WSe_2}$). **c** (**e**) Top: Energy and FWHM of each Lorentzian peak obtained from the fitting procedure as shown in (**b**). **d**. Bottom: Amplitude of each Lorentzian peak obtained from the fitting procedure as shown in (**b**). **d** The red and blue stars indicate $A_{1s}^{WSe_2}$ and $A_{1s}^{MoSe_2}$, respectively.

transport direction being forbidden by the junction, in agreement with[18,28]. The difference in the local dielectric environment between e-LH and un-LH has no impact on the diode-like effect. There is no visible influence on both the energy and FWHM of the PL spectra after crossing or being blocked at the junction. This suggests that

there is no alteration of the nature of the excitons in each TMD-ML near the vicinity of the junction.

It is very interesting to notice that, in both e-LH and un-LH, the TEPL intensity originating from WSe₂ (red stars) strongly decreases as the excitation takes place closer to the interface ($x_{tip} \rightarrow 0$). As a

consequence, the resulting signal measured near the LH interface is strongly dominated by $A_{1s}^{MoSe_2}$ (see Fig. 2c, e bottom panels). In other words, even if the tip is located inside WSe₂, most of the generated excitons migrate through the LH junction into MoSe₂ before recombining. This behavior can only be explained by a strongly anisotropic transport, with the diffusion toward the junction becoming more efficient than the other directions. The second effect resulting from such an efficient exciton transfer $A_{1s}^{WSe_2} \rightarrow A_{1s}^{MoSe_2}$ is that as the tip approaches the interface from the WSe₂ side the MoSe₂ PL intensity $I_{PL}(A_{1s}^{MoSe_2})$ increases and nearly reaches the values of the integrated PL intensity of WSe₂ far from the interface ($I_{PL}^{\infty}(A_{1s}^{WSe_2})$). Far from the junction, MoSe₂ is three times less bright than WSe₂, meaning that the junction is enhancing the MoSe₂ flake brightness at its vicinity.

Finally, a pronounced difference can be seen between the two systems, with $A_{MoSe_2}^{1s}$ signature extending ~1.2 $\mu m$ away from the interface in un-LH versus ~400 nm for e-LH. Both distances being considerably larger than the physical junction width (≈3 nm[18])

## Modified exciton transfer model

In order to get a better understanding of the underlying physical phenomena taking place close to the interface, we developed an exciton transfer model, which is detailed in supplementary information. We theoretically investigate, in the low exciton density regime (typically $10^{11}$ cm⁻²), the variation of neutral exciton density versus tip position, to be compared to the TEPL results. To do so, we analytically solved the linear 1D steady-state diffusion equation in each material $i$ (1 = WSe₂, 2 = MoSe₂), given by :

$$D_i \frac{d^2 n(x, x_{tip})}{dx^2} - \frac{n(x, x_{tip})}{\tau_i} + \Gamma_i(x, x_{tip}) = 0 \qquad (1)$$

$n(x, x_{tip})$ is the exciton density at $x$ position with the excitation taking place at $x_{tip}$, the $x$-axis origin being placed at the LH interface, $D_i$ the effective diffusion coefficient and $\tau_i$ the effective lifetime, radiative ($\tau_i^r$) and non-radiative ($\tau_i^{nr}$) ($1/\tau_i = 1/\tau_i^r + 1/\tau_i^{nr}$). $L_i = \sqrt{D_i \tau_i}$ represents the effective diffusion length in the material $i$. We use $\Gamma_i(x, x_{tip}) = \Gamma_{0_i}(P, \alpha_i, \nu) e^{-\frac{(x-x_{tip})^2}{w^2}}$ to simulate the exciton generation under the tip, centered at $x_{tip}$. The width $w$ corresponds to the tip diameter (~30 nm), $P$ is the enhanced laser power under the tip and $\alpha_i$ the absorption coefficient of the material $i$ at the laser energy $h\nu$. We set boundary conditions to be $n(x \rightarrow \pm\infty, x_{tip}) = 0$. In agreement with experimental results, we impose the continuity of the integrated PL intensity for all values of $x_{tip}$. Finally, we considered that bright excitons are not interacting with other types of excitons (dark excitons, B-excitons, momentum-forbidden excitons).

As illustrated in Fig. 3a, we model the junction as an ideally thin interface of fixed width ($\epsilon$) much smaller than the excitonic diffusion length inside the barrier ($\ll L_D$), where no electron-hole pair recombination can occur. To model the asymmetric local effective drift of neutral exciton through the junction, we introduce a local uniform force field:

$$\vec{F} = -\vec{\nabla} E_{A_{1s}} \simeq -\left(E\left(A_{1s}^{WSe_2}\right) - E\left(A_{1s}^{MoSe_2}\right)\right)/\epsilon \ \hat{x} \qquad (2)$$

As a result, the junction imposes the continuity of the excitonic flux density on both sides of the interface by a local constant flux density $\vec{j}_n = \mu_b \vec{F} n - D_b \vec{\nabla} n$, where $\mu_b$ and $D_b$ are the exciton mobility and diffusion coefficient inside the interface.

Figure 3b displays the calculated $n(x, x_{tip})$ for two different tip positions. We emphasize first on the lateral heterostructure specificity: the exciton density is discontinuous at the interface (see continuous lines). As a reference, the continuous exciton density calculated for classical diffusion ($F = 0$) is display in dashed lines. When the tip

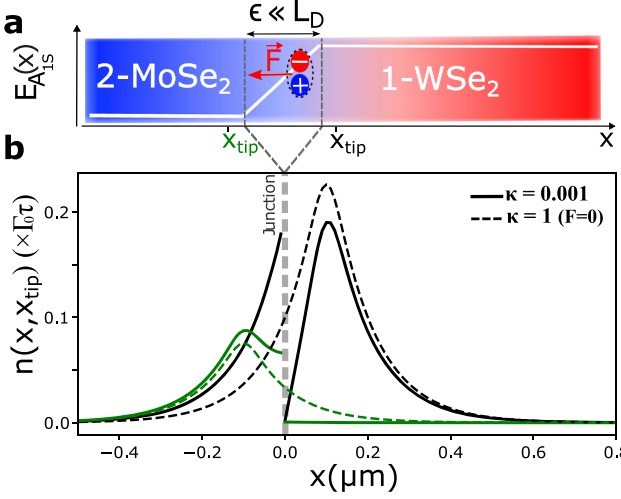

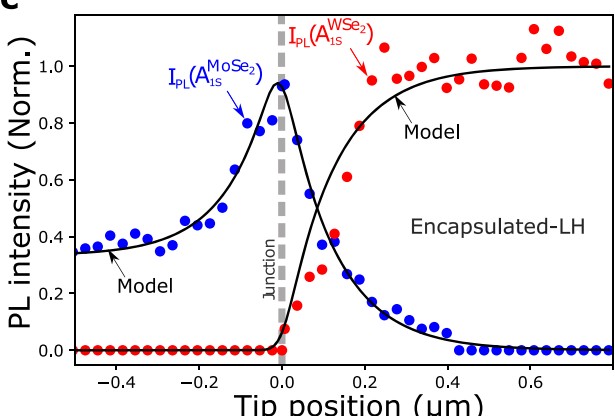

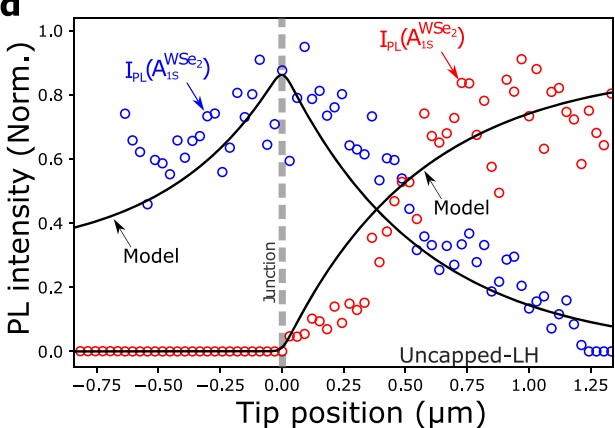

**Fig. 3 | Exciton distribution and near field optical spectroscopy of the junction. a** Illustration of the exciton drift inside the interface, the shaded area corresponds to the partition zone between the two materials. **b** Exciton density $n(x, x_{tip})$ calculated with the near-field model for two tip positions $x_{tip} = L_D$ (black line) and $x_{tip} = -L_D$ (green line) with: $\Gamma_{0_1} = 3\Gamma_{0_2} = \Gamma_0$, $\tau_1 = \tau_2 = \tau$, $L_1 = L_2 = L_D = 0.1 \mu m$. The dashed lines indicate the exciton density with no junction influence ($F = 0$). **c** Normalized PL intensity of $A_{1s}^{MoSe_2}$ (blue) and $A_{1s}^{WSe_2}$ (red) in e-LH area. **d** Same as (**c**) in the un-LH area. The solid black line represents the PL intensities calculated using Eqs. (5) and (6).

excitation takes place inside MoSe₂, the excitons are blocked by the junction, and the exciton density on the other side of the junction stays close to zero (see continuous green line). Excitons interacting with the junction loose a velocity $v^* = \mu_b F$ blocking the diffusion process. This blocking phenomenon causes a redistribution of the exciton that

strongly accumulate at the interface. On the other hand, when the excitation takes place inside WSe$_2$, the excitons cross the junction, and are accelerated, hence acquiring an extra velocity $v^*$. One can clearly see that this drives those excitons further inside the MoSe$_2$ layer, while bringing the excitonic density down to zero near the interface on the WSe$_2$ side (see continuous black line).

To facilitate further developments, and better characterize the exciton discontinuity at junction we define a partition coefficient linking the exciton densities $n(x=0^+)$ and $n(x=0^-)$ on both sides of the interface. Because of the asymmetrical behavior of the junction, this partition coefficient $\kappa = n(x=0^+)/n(x=0^-)$ can be written, depending on the direction of excitonic flux, as follows:

$$\kappa \simeq \begin{cases} \kappa^+ = \left(1 - \frac{L_2}{\mu_b F \tau_2}\right) e^{-\frac{\mu_b F \epsilon}{D_b}} + \frac{L_2}{\mu_b F \tau_2} & x_{\text{tip}} > 0 \\ \kappa^- = \left[\left(1 + \frac{L_1}{\mu_b F \tau_1}\right) e^{\frac{\mu_b F \epsilon}{D_b}} - \frac{L_1}{\mu_b F \tau_1}\right]^{-1} & x_{\text{tip}} < 0 \end{cases} \quad (3)$$

with $L_1$, $L_2$ the diffusion lengths of both materials. $\kappa$ quantifies the discontinuity of the exciton density at the junction, with $\kappa^+ = \kappa(x_{\text{tip}} > 0)$ describing the exciton diffusion from WSe$_2$ to MoSe$_2$ and $\kappa^- = \kappa(x_{\text{tip}} < 0)$ the diffusion from MoSe$_2$ to WSe$_2$. In our case, as the local equilibrium is established in the steady state, $\kappa^+ \sim \kappa^-$ (see Table 1), to simplify the discussion we will therefore refer simply to $\kappa$ independently of the direction of excitonic flux. When $\kappa = 1$, the exciton distribution, displayed as a dashed line in Fig. 3b, is continuous as observed in classical diffusion. When $\kappa \ll 1$, corresponding to the present case, the exciton distribution is strongly asymmetric (see continuous line in Fig. 3b). Our finding can be interpreted as an extrapolation of the interfacial thermal resistance so-called Kapitza resistance, which describes the temperature discontinuity at atomically flat interface between two materials[29,30]. For high-quality LHs, the absolute value of the exciton Kapitza resistance can be defined as a function of the partition coefficient as follows: (See more details in supplementary information).

$$R_n = \frac{n(x=0^-) - n(x=0^+)}{j_n} \simeq \frac{\kappa - 1}{v^*} \frac{1 - e^{v^* \epsilon/D_b}}{1 - \kappa e^{v^* \epsilon/D_b}} \quad (4)$$

To compare directly the prediction of our near-field model with the experimental results, we calculate $I_{\text{PL}}(A_{1s}^{\text{MoSe}_2})$ and $I_{\text{PL}}(A_{1s}^{\text{WSe}_2})$, the integrated PL intensities of each material as a function of the tip position. In the linear regime, the normalized PL intensity of MoSe$_2$ and WSe$_2$ can be written as (see supplementary information for details):

$$I_{\text{PL}}^{\text{Norm}}\left(A_{1s}^{\text{MoSe}_2}\right)(x_{\text{tip}}) = \frac{A_\kappa}{\sqrt{\pi}w} \int_0^\infty dx e^{-x/L_1} e^{-\frac{(x-x_{\text{tip}})^2}{w^2}}$$
$$+ \frac{1}{\sqrt{\pi}\beta w} \int_{-\infty}^0 dx \left(1 + (\beta A_\kappa - 1) e^{x/L_2}\right) e^{-\frac{(x-x_{\text{tip}})^2}{w^2}} \quad (5)$$

$$I_{\text{PL}}^{\text{Norm}}\left(A_{1s}^{\text{WSe}_2}\right)(x_{\text{tip}}) = \frac{1}{\sqrt{\pi}w} \int_0^\infty dx \left(1 - (1 - B_\kappa) e^{-x/L_1}\right) e^{-\frac{(x-x_{\text{tip}})^2}{w^2}}$$
$$+ \frac{B_\kappa}{\sqrt{\pi}w} \int_{-\infty}^0 dx e^{x/L_2} e^{-\frac{(x-x_{\text{tip}})^2}{w^2}} \quad (6)$$

with $I_{\text{PL}}^\infty(A_{1s}^{\text{WSe}_2})$ used for normalization. The experimental PL is fitted using four fitting parameters: $A_\kappa$ and $B_\kappa$, amplitude parameters, giving us access to $\kappa$, the partition coefficient and $L_1$, $L_2$. Figure 3c, d shows the experimental normalized PL intensities of MoSe$_2$ (blue dots) and WSe$_2$ (red dots) for both e-LH and un-LH configurations, respectively. The black solid lines represent the fits using Eqs. (5) and (6). The resulting fitting parameters are displayed in Table 1. In both systems, the $\kappa \ll 1$ values for e-LH and un-LH confirm the strong asymmetry of the junction and the resulting strong Kapitza exciton resistance.

## DISCUSSION

One can see that the model describes extremely well the strong quenching of the WSe$_2$-related PL and the appearance of MoSe$_2$-related PL when approaching the junction from the right ($x_{\text{tip}} > 0$), shedding light on the experimental results on unidirectional excitonic transport across the junction. It is also able to quantitatively describe the enhancement of $I_{\text{PL}}(A_{1s}^{\text{MoSe}_2})$ at the junction and its progressive decrease to the typically lower value $I_{\text{PL}}^\infty(A_{1s}^{\text{MoSe}_2})$ of monolayer MoSe$_2$ as the tip is moved away from the junction ($x_{\text{tip}} < 0$). We would like to point out that this observation is far from trivial, it indicates that to conserve the experimentally observed continuity of the PL intensity, the exciton density at the interface ($n(x=0^\pm)$) is enhanced. We believe that this could be linked to the drastic change in the exciton velocity at the interface. Indeed, when the tip is inside MoSe$_2$, excitons diffusing toward WSe$_2$ are abruptly stopped at the junction, their average velocity going to zero. The non-radiative lifetime ($\tau_2^{nr}$), being in part linked to the probability for the exciton to encounter non-radiative traps during its lifetime, is by consequence increased[31], resulting in an increased population at the interface. This would explain why the size of PL enhancement area (where $I_{\text{PL}}(A_{1s}^{\text{MoSe}_2}) \geq I_{\text{PL}}^\infty(A_{1s}^{\text{MoSe}_2})$) is a function of the diffusion length (see Fig. 3), as it only occurs when excitons start accumulating at the junction. Other phenomenon may contribute to this variation as a modification of the non-radiative Auger recombination due to the local enhancement of the exciton density.

Finally, the model reveals that the important difference observed in the PL profile between e-LH and un-LH is linked to drastically different values of the diffusion length, much shorter in e-LH than in un-LH (see experimental results in Fig. 2 and theoretical results in Table 1). To understand why $L_D^{e-LH} < L_D^{un-LH}$ we need to consider that even if the TEPL experiments on e-LH and un-LH were performed using the same laser power ($400\mu$W), the presence or absence of the top hBN layer, strongly impacts the near-field configuration under the tip. In first approximation, and considering a spherical tip of 30nm diameter, we can estimate the near-field intensity ratio on the TMD layer as:

$$\frac{\left|\vec{E}\right|^2_{un-LH}}{\left|\vec{E}\right|^2_{e-LH}} \simeq \left(\frac{Z_{e-LH}}{Z_{un-LH}}\right)^6 \left(\frac{n_{BN}}{n_{air}}\right)^4 \quad (7)$$

$Z_{e-LH}$ and $Z_{un-LH}$ are the distances from the tip center to the TMD-ML in e-LH and un-LH areas, respectively. $n_{BN}$ and $n_{air}$ are hBN and air refractive index, respectively. Considering a 3 nm-thick hBN top layer (measured by AFM), we can estimate the enhancement ratio to be ~23. We can then expect the exciton density generated under the tip in un-LH to be more than one order of magnitude larger than in e-LH. To characterize the exciton transport properties of the WSe$_2$ layer versus exciton density we use spatio-temporal PL signal in a confocal microscope. A local pulsed laser excitation ($\lambda$= 692 nm, pulse duration 1.5 ps), generates excitons, and we record the time evolution of the PL profile. We then extract the time evolution of the squared width $w^2(t) = w^2(0) + \Delta w^2(t)$ which is used to determine an effective diffusion coefficient in the 80 first picosecond where 90% of the PL signal originates : $\Delta w^2(t) = 4D_{\text{eff}}t$. An important point is to be able to decorrelate the effective lifetime $\tau_{eff}$ (linked to the PL decay) and the

**Table 1 | Results of the PL fits using the near-field model for both e-LH and un-LH systems**

| Area | $L_1$(nm) | $L_2$(nm) | $\kappa$ | $R_n$(s m$^{-1}$) |
|------|-----------|-----------|----------|--------------------|
| e-LH | 120 ± 6 | 110 ± 5.5 | ~1 × 10$^{-3}$ | ~1 × 10$^{-5}$ |
| un-LH | 550 ± 55 | 450 ± 45 | ~1 × 10$^{-3}$ | ~1 × 10$^{-5}$ |

diffusion coefficients, from which we deduce the effective diffusion length $L_{eff} = \sqrt{D_{eff}\tau_{eff}}$. Experimental conditions and detailed results are given in the supplementary information (Figs. 2–4). The excitation power density was varied over several order of magnitude and the results compared between the e-LH and un-LH samples. Figure 4 shows that, for both samples, $L_{eff}$ increases with the excitation power density (i.e. the excitonic density) which is mainly due to the increase of the effective diffusion coefficient (See supplementary information). This trend, attributed to a weak Auger contribution has been previously observed in $WS_2$[31,32] and explain the large difference observed in the diffusion lengths we observed in e-LH and un-LH. The experimental results have been modeled by using a Auger coefficient of $0.2\,10^{-6}$ cm$^2$/s (See supplementary information). This value is two orders of magnitude lower than the one estimated on non encapsulated $WS_2$ (without top and bottom hBN) where a large difference of the transport properties values with a fully encapsulated layer was observed[31]. In summary, we show that both samples exhibit similar exciton transport properties, indicating the importance of the hBN bottom layer that prevents strong Auger scattering effect.

One can notice here that, according to Eq. (7), the enhancement ratio is dominated by the tip-TMD layer distance. This offers a unique opportunity, as using plasmonic tips or nanoantennas in combination with hBN encapsulation allows to control the diffusion length in this type of structures. As a matter of fact, controlling the thickness of the top hBN layer from a single layer to 20 layers, for example, would change the enhancement factor by a factor 10, modifying the diffusion length by the same amount, all the while avoiding the flaws, lower optical quality and exposition to environment, that are observed in the uncapped LH.

In summary, we have performed a detailed tip-enhanced spectroscopy study of a $MoSe_2$-$WSe_2$ lateral heterostructure, and have developed a model to render the observation of the unilateral transport across the junction observed in near-field PL experiment. It accounts for a discontinuity of the exciton density at the interface. We have thus shown that the exciton diffusion properties follow a semi-classical process, due to the difference in the energy gap between the two materials: the usual isotropic in-plane diffusion of the excitons is frustrated by the junction, leading to an asymmetric diffusion, in which all generated excitons move away from the high bandgap $WSe_2$ layer to

recombine in the low bandgap $MoSe_2$ layer. This transfer causes near the interface the quenching of $WSe_2$-related PL and the enhancement of the $MoSe_2$-related PL, well above the one observed in "bulk" $MoSe_2$ far from the junction. Interestingly, we observe similar asymmetric diffusion property for samples with or without top hBN. Both samples present similar intrinsic transport properties, probably linked to an efficient screening of the dielectric disorder by the bottom encapsulation by hBN layer. Finally, we have shown that the diffusion length in $WSe_2$ is strongly dependent on the exciton density. This offers a new degree of freedom, as changing the laser power density or the near-field enhancement (for instance using optically resonant nanoantennas instead of the plasmonic tip) would allow tuning the diffusion length to any wanted value from tens of nanometer up to few micrometers. TMD-based lateral heterostructure is a rapidly evolving research field that could offer to combine TMDs with very different bandgaps, thus allowing partition zone engineering, or creating more complex designs using three or more TMDs (lateral excitonic quantum wells). This work offers both the theoretical and experimental tools to predict and control the new diffusion properties that will be at the origin of new excitronic devices.

## Methods

We use water-assisted deterministic transfer to pick up the chemical vapor deposition LH from the as-grown substrate using polydimethylsiloxane (PDMS) and transfer it on a supporting flake of exfoliated hBN on $SiO_2$/Si substrate[26]. TERS and TEPL are carried out with state-of-the-art commercial system (Trios OmegaScope-R coupled with LabRAM spectrometer, Horiba Scientific). Silver-coated tips with an apex radius of 15nm were used for tip-enhanced measurements. The Time evolution of the PL profile experiment is based on a diffraction-limited laser excitation that induces lateral diffusion of the photogenerated excitonic species. We used a Streak camera system to record the time evolution of the PL spatial profile $I_{PL} = I(x, t)$ with a time resolution of 5.5 ps. The Ti:Sa laser excitation is set to $E_{ex} = 1.79$ eV, with a 80 MHz repetition frequency, 1.5 ps pulse width and we vary the excitation power from 10 μW to 1 mW.

## Data availability

The data for all figures in the main text are available in the Source Data file. The data presented in Supplementary information are available upon request due to their large file size. Source data are provided with this paper.

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

## Acknowledgements

Toulouse acknowledges partial funding from ANR IXTASE (ANR-20-CE30-0032), ANR HiLight (ANR-19-CE24-0020-01), ANR Ti-P (ANR-21-CE30-0042), NanoX project 2DLight (ANR-17-EURE-0009), the Institut Universitaire de France, and the EUR grant ATRAP-2D NanoX ANR-17-EURE-0009 in the framework of the "Programme des Investissements d'Avenir", the Institute of quantum technology in Occitanie IQO and a UPS excellence PhD grant. Growth of hexagonal boron nitride crystals was supported by JSPS KAKENHI (Grants No. 19H05790, No. 20H00354 and No. 21H05233). The Jena group received financial support of the Deutsche Forschungsgemeinschaft (DFG) through a research infrastructure grant INST 275/257-1 FUGG, CRC 1375 NOA (Project B2), SPP2244 (Project TU149/13-1) as well as DFG grant TU149/16-1. This project has also received funding from the joint European Union's Horizon 2020 and DFG research and innovation program FLAG-ERA under grant TU149/9-1.

## Author contributions

Z.G., A.G., and A.T. developed the growth method and fabricated the lateral heterostructures. T.T. and K.W. grew the hBN crystals. I.P. carried out hBN encapsulation. D.B., L.L., P.R, D.L., and X.M. developed, performed and analyzed effective diffusion coefficient and exciton lifetime experiments. H.L. and J.-M.P. performed TEPL and TERS. V.P. and J.-M.P. optimized the TEPL and TERS set-up. H.L., J.-M.P., N.C, and V.P developed the analytical modeling. J-M.P., V.P., L.L., B.U. wrote the manuscript with inputs from all the authors and supervised the project.

## Competing interests

The authors declare no competing interests.
