## [Peer Review File · Nature Communications]

Reviewers' Comments:

Reviewer #1:

Remarks to the Author:

Comments on "Kapitza-resistance-like exciton dynamics in atomically flat MoSe₂-WSe₂ lateral heterojunction"

Lamsaadi and coworkers present a nanoscale study of exciton diffusion processes across a lateral interface between monolayer (ML) MoSe₂ and WSe₂ TMDs, which form a lateral heterostructure (LH). As is well explained in the abstract, exciton diffusion in TMD MLs is of intense general interest given that TMDs MLs and their heterostructures (HS) are very promising materials for a variety of optoelectronic and spintronic device applications due to their optoelectronic properties. Here, the authors study the diffusion of excitons excited on either side of the MoSe₂-WSe₂ interface using AFM-tip enhanced excitation (~30 nm diameter excitation profile), well above the bandgap of both materials, and then detect emitted PL with a large collection angle. The authors study two regions of the same sample; one which is fully encapsulated in hBN (e-LH) and one which is only partially encapsulated (no top layer of hBN, un-LH). The presented manuscript presents two particularly significant findings in the development of exciton-based electronics: the primary observations are of marked asymmetry in the PL as a function of excitation position which is interpreted as discontinuous exciton diffusion at the HS interface, described in analogy to thermal Kapitza-resistance. The authors construct a model of the observed exciton diffusion which fits the observations well. The second significant finding is that the exciton diffusion lengths are significantly (~4x) larger in the un-LH than the e-LH; this is attributed to the fact that the e-LH sample is offset from the excitation tip by the hBN thickness, which results in a lower overall excitation density and therefore lower diffusion length. The authors suggest that hBN layer thickness could be used as an additional control parameter in the design and implementation of devices which rely on exciton diffusion.

The data presented are high quality, with 30 nm spatial resolution steady state and time-resolved tip-enhanced PL measurements of a micron-scale sample. The qualitative interpretation of the results is compelling, though the theoretical model requires more context and neglects to consider excitation of higher-energy MoSe₂ B excitons. It appears that the manuscript presented here is an expansion of previously published work by the same authors recently published in npj 2D materials and applications (reference 16) such that the authors should make it clear how this manuscript is distinct from the findings in reference 16.

There are several significant issues that require attention prior to publication, detailed below, and I recommend the manuscript be reconsidered following revisions.

Major revisions:

- A previous study by the authors (ref. 16) presents similar data (ref. 16 Fig. 3) and theoretical modeling (ref. 16, Fig 4) as in the present manuscript (compare to Fig 2 & 3). It is clear that the manuscript under review expands upon the previous work (e.g. additionally studying the un-LH region and expanding the modeling) and that the previous work describes the initial characterization of the sample (e.g. junction location & width). More discussion is needed to put the current work in the context of this previous study, in particular how this work is distinct from and builds upon that in ref. 16.
- Though I am not intimately familiar with exciton transfer models, more background and description are needed for the theory introduced in section III. There are very few references to previous work on exciton transfer models in this section despite many existing in the literature. Please provide a discussion of how the presented model and approach developed in this work compare to other studies of exciton diffusion at interfaces. Please consider including reference to works on exciton diffusion by Libai Huang [DOI: 10.1021/acs.jpcclett.7b00885] and Parag Deotare [DOI: 10.1063/1.5063263, 10.1021/acsami.8b12291] in particular.
- The electronic properties of the two MLs do not appear to enter into the description of the exciton diffusion beyond the definition of the force field in equation 2, which essentially describes an exciton energy difference. The pump energy ($\hbar\omega = 1.96$ eV) is well above the bandgap of both MoSe₂ and WSe₂ A excitons ($\hbar\omega \sim 1.55, 1.65$ eV, respectively) and could reasonably excite higher energy B excitons in the MoSe₂ layer ($\hbar\omega \sim 1.78$ eV). Given the model presented in the

manuscript, excitation of MoSe₂ B excitons could transfer to the WSe₂ layer given the energetic favorability. This is not supported by the experimental findings and suggests that the energetic funnel description (equation 2) is insufficient for describing the presented results. Were higher energy excitons considered in the model of exciton diffusion? Please provide a discussion of MoSe₂ B exciton excitation in the present study and an explanation for why they can be neglected.

- In the definition of the partition coefficient, κ , in equation 3 it appears the same exciton mobility and diffusion coefficients are used for both materials. Please provide explanation of why the same coefficients can be used for both materials given differences in electronic and structural properties.
- Given the importance of the junction location to the interpretation of the data (e.g. the definition of $X_{\text{tip}} = 0$ in Fig. 2), the authors should provide further description of how the junction location is determined with tip enhanced Raman spectroscopy. It would be illustrative if the authors could provide a supplementary figure to this effect.

Minor revisions:

- Perhaps I am misunderstanding the meaning, but it seems that the position-dependent partition functions, κ_+ and κ_- , are dispensed with on line 193-194. What is the reason for defining the discontinuous partition function for different spatial regions in equation 3 if it is then immediately disregarded (“...we will refer simply to κ independently of the source position”)?
- Given the prominence of the analogy to thermal Kapitza resistance in the title, I suggest introducing the analogy in the introduction (perhaps after line 56) rather than on page 10.
- To better illustrate the exciton diffusion process taking place, I suggest creating a diagram of the different excitonic processes taking place when excitation occurs on either side of the junction.
- In the discussion of the spatially-averaged PL starting on line 87, the authors describe how the PL spectra of the un-LH differs from the e-LH spectra, including increased peak widths, intensity, and asymmetry. Given the number of differences between the two sample regions, I suggest removing the phrase “nearly identical” on line 88.
- The cause of the increased intensity in the un-LH sample region is initially surprising (line 89) given the lack of the capping hBN and is not discussed in detail until page 12; I suggest adding a statement alluding to this detailed discussion on page 12 in the initial description on page 4.
- An exciting proposal from this study is the ability to control exciton diffusion lengths by deposition of different thicknesses of hBN (or similar materials). Is it possible to replicate the exciton diffusion behavior of the e-LH region in the un-LH by displacing the AFM tip from the surface?
- On line 248 and in the discussion of the time-resolved diffusion profile that follows, please include references to the corresponding figures and discussion in the Supplementary text.
- The PL spectra in Figure 1 C & F are referred to as “typical far-field PL;” please clarify how this is different than the PL TEPL in Figure 2 and beyond.
- Please provide a brief description or reference as to how the AFM tip-enhanced Gaussian excitation profile determined (line 109).
- When describing the experimental setup, please also provide the pump laser energy (~ 1.96 eV) for better comparison with the PL data presented in eV in the manuscript figures.
- Please define ϵ and LD in their first usage on line 171.
- Please define L1 and L2 after equation 3 (line 191) rather than line 209.
- Many of the references show an error where the titles are immediately preceded by “en.”
- Throughout the manuscript, the supplementary information is referred to in different ways.

Reviewer #2:

Remarks to the Author:

The manuscript by Lamsaadi et al. presents an interesting study of exciton transport across a lateral TMD heterojunction. The authors use tip-enhanced photoluminescence spectroscopy to study the propagation of photo-excited excitons and draw conclusions from changes in the luminescence spectrum. To complement these measurements, the authors use a classical drift-diffusion model to numerically analyze the experimental behavior. Using reasonable values for input parameters, the authors achieve good agreement between the numerical and experimental results.

Both the lateral heterostructure and the experimental technique employed in this study provide a

new perspective on exciton transport in TMD-based structures. The experiments and simulations support the main claims of the manuscript. Thus I believe this manuscript will be of interest to a broader community of researchers working on exciton transport. Therefore I support the publication of this manuscript.

I have a few minor comments:

The lack of hBN layers on top of the heterostructure is presented as an advantage, because it results in higher exciton densities created by the optical near-field excitation. I must say that I am not very fond of this argumentation, since it suggests that inherently changes when one does not use top hBN. As the authors discuss, however, the density-dependent increase in the exciton diffusion coefficient is merely due to effective changes resulting from non-radiative exciton recombination in high-density regions of the excitation area. To justify the claim that the lack of top-hBN actually enhances exciton propagation, the authors would need to present a quantitative comparison of PL intensities observed in the different regions of their e-LH and un-LH samples. They should show that the PL intensity at the interface is indeed larger when the "effective" diffusion coefficient has increased, which requires a careful study of the excitation power-dependence.

The authors argue that the slow-down of excitons at the interface leads to local increases in the exciton density and the non-radiative lifetime, which explains the observed increase in PL intensity. However, one could argue that non-radiative Auger recombination also increases due to the local enhancement of the exciton density. Maybe the authors can include this point in their discussion.

Reviewer #3:

Remarks to the Author:

In this manuscript, the authors described a Kapitza-like-Resistabnce exction dynamics in a 2D heterogeneous structure. The results reported an excellent data on the unidirectional excitonic transport, showing a huge progress in this field. In addition, a theoretical model is shown in this work to approval this experimental results. After reviewing, in my opinion, it could be considered acceptable in Nature Communications. Some suggestions are provided for authors to consider:

1. In the introduction part, the authors should state how to control the exciton in BN?
2. In Fig. 1a, the authors have to distinguish the heterogeneous region of the two materials.
3. The color distribution in Figure 1C&G is not uniform, and the intensity varies too much, how to prove that the test area, which truly reflects the material properties?
4. In line 78, dashed blue should be dashed white?
5. In addition, the authors should add some discussion on the PL spectra, in order to show more clear to readers.

REVIEWER COMMENTS

We thank all the referees for their positive and very detailed constructive review. All the modified parts of the text following the referees' comments will appear in red.

Reviewer #1 (Remarks to the Author):

Comments on "Kapitza-resistance-like exciton dynamics in atomically flat MoSe₂-WSe₂ lateral heterojunction"

Lamsaadi and coworkers present a nanoscale study of exciton diffusion processes across a lateral interface between monolayer (ML) MoSe₂ and WSe₂ TMDs, which form a lateral heterostructure (LH). As is well explained in the abstract, exciton diffusion in TMD MLs is of intense general interest given that TMDs MLs and their heterostructures (HS) are very promising materials for a variety of optoelectronic and spintronic device applications due to their optoelectronic properties. Here, the authors study the diffusion of excitons excited on either side of the MoSe₂-WSe₂ interface using AFM-tip enhanced excitation (~30 nm diameter excitation profile), well above the bandgap of both materials, and then detect emitted PL with a large collection angle. The authors study two regions of the same sample; one which is fully encapsulated in hBN (e-LH) and one which is only partially encapsulated (no top layer of hBN, un-LH). The presented manuscript presents two particularly significant findings in the development of exciton-based electronics: the primary observations are of marked asymmetry in the PL as a function of excitation position which is interpreted as discontinuous exciton diffusion at the HS interface, described in analogy to thermal Kapitza-resistance. The authors construct a model of the observed exciton diffusion which fits the observations well. The second significant finding is that the exciton diffusion lengths are significantly (~4x) larger in the un-LH than the e-LH; this is attributed to the fact that the e-LH sample is offset from the excitation tip by the hBN thickness, which results in a lower overall excitation density and therefore lower diffusion length. The authors suggest that hBN layer thickness could be used as an additional control parameter in the design and implementation of devices which rely on exciton diffusion.

The data presented are high quality, with 30 nm spatial resolution steady state and time-resolved tip-enhanced PL measurements of a micron-scale sample. The qualitative interpretation of the results is compelling, though the theoretical model requires more context and neglects to consider excitation of higher-energy MoSe₂ B excitons. It appears that the manuscript presented here is an expansion of previously published work by the same authors recently published in npj 2D materials and applications (reference 16) such that the authors should make it clear how this manuscript is distinct from the findings in reference 16.

There are several significant issues that require attention prior to publication, detailed below, and I recommend the manuscript be reconsidered following revisions.

Major revisions:

- A previous study by the authors (ref. 16) presents similar data (ref. 16 Fig. 3) and theoretical modeling (ref. 16, Fig 4) as in the present manuscript (compare to Fig 2 & 3). It is clear that the manuscript under review expands upon the previous work (e.g. additionally studying the un-LH region and expanding the modeling) and that the previous work describes the initial characterization of the sample (e.g. junction location & width). More discussion is needed to put the current work in the context of this previous study, in particular how this work is distinct from and builds upon that in ref. 16.

To enlighten the connection between our previous work and the present one, the text has been modified. The following sentences have been added to the introduction to highlight the novelty of the present work (line 50): *"With the increasing maturity of the field, several basic components necessary to control exciton information have been developed, like room temperature excitonic transistor in a Van der Waals vertical heterostructure [14], or an excitonic diode able to filter excitons in lateral heterostructures [15–17]. The physical mechanisms involved in the exciton transfer processes through lateral heterojunctions, as well as their influence on the exciton distribution, dynamics and the resulting photoluminescence observed in previous work [15] are still unclear. This is mainly due to: (i) the extreme sharpness of lateral heterojunctions making complex its optical characterization, (ii) the strong influence of external parameters like defects, strain and dielectric environment on exciton transport, making the fabrication of high-quality samples with controlled dielectric environment crucial to extract the intrinsic junction properties, (iii) the lack of a complete theoretical description able to describe the asymmetric exciton transfer through the junction."*

- Though I am not intimately familiar with exciton transfer models, more background and description are needed for the theory introduced in section III. There are very few references to previous work on exciton transfer models in this section despite many existing in the literature. Please provide a discussion of how the presented model and

approach developed in this work compare to other studies of exciton diffusion at interfaces. Please consider including reference to works on exciton diffusion by Libai Huang [DOI: 10.1021/acs.jpcclett.7b00885] and Parag Deotare [DOI: 10.1063/1.5063263, 10.1021/acsami.8b12291] in particular.

The referee is perfectly right, to extend the physical background the text has been modified with the following sentences added to the introduction of the paper (line 61): “Exciton diffusion at sharp interfaces have been previously studied in other materials like organic semiconductors in so-called bulk-heterojunction [18, 19]. In the context of solar cells, efficient fast exciton diffusion and harvesting have stimulated strong theoretical and experimental efforts. Nevertheless, in such system the interface is designed to dissociate the exciton into free charges and not to transfer the entire quasi-particle through the junction [18, 19]”

[18] Y. Firdaus, V. M. Le Corre, S. Karuthedath, W. Liu, A. Markina, W. Huang, S. Chattopadhyay, M. M. Nahid, M. I. Nugraha, Y. Lin, A. Seithkan, A. Basu, W. Zhang, I. McCulloch, H. Ade, J. Labram, F. Laquai, D. Andrienko, L. J. A. Koster, and T. D. Anthopoulos, *enLong-range exciton diffusion in molecular non-fullerene acceptors*, *Nature Communications* 11, 5220 (2020).

[19] B. Siegmund, M. T. Sajjad, J. Widmer, D. Ray, C. Koerner, M. Riede, K. Leo, I. D. W. Samuel, and K. Vandewal, *enExciton Diffusion Length and Charge Extraction Yield in Organic Bilayer Solar Cells*, *Advanced Materials* 29, 1604424 (2017).

We thank the referee for the suggested references, that have been added to the text.

[10] L. Yuan, T. Wang, T. Zhu, M. Zhou, and L. Huang, *Exciton Dynamics, Transport, and Annihilation in Atomically Thin Two-Dimensional Semiconductors*, *The Journal of Physical Chemistry Letters* 8, 3371 (2017).

[23] D. F. Cordovilla Leon, Z. Li, S. W. Jang, C.-H. Cheng, and P. B. Deotare, *Exciton transport in strained monolayer WSe₂*, *Applied Physics Letters* 113, 252101 (2018).

- The electronic properties of the two MLs do not appear to enter into the description of the exciton diffusion beyond the definition of the force field in equation 2, which essentially describes an exciton energy difference. The pump energy ($\hbar\omega = 1.96$ eV) is well above the bandgap of both MoSe₂ and WSe₂ A excitons ($\hbar\omega \sim 1.55, 1.65$ eV, respectively) and could reasonably excite higher energy B excitons in the MoSe₂ layer ($\hbar\omega \sim 1.78$ eV). Given the model presented in the manuscript, excitation of MoSe₂ B excitons could transfer to the WSe₂ layer given the energetic favorability. This is not supported by the experimental findings and suggests that the energetic funnel description (equation 2) is insufficient for describing the presented results. Were higher energy excitons considered in the model of exciton diffusion? Please provide a discussion of MoSe₂ B exciton excitation in the present study and an explanation for why they can be neglected.

The B-exciton peak is associated with transitions between the spin-orbit split valence band and the conduction band. In the photoluminescence spectra of our CVD-grown MoSe₂ and WSe₂ monolayers, and in agreement with the literature [Kathleen M. McCreary, Aubrey T. Hanbicki, Saujan V. Sivaram, Berend T. Jonker; A- and B-exciton photoluminescence intensity ratio as a measure of sample quality for transition metal dichalcogenide monolayers. *APL Mater* 1 November 2018; 6 (11): 111106] the B-exciton peak appears at least two orders of magnitude lower in amplitude than the dominant A-peak. This suggests that the B-exciton population is very low compare to the population of A-exciton with B-exciton lifetime been considerably shorter due to the rapid relaxation to the A-exciton state much more favorable as it corresponds to the radiative recombination via the lowest energy channel. Nevertheless, should B-exciton contribution become intense for any reason, the experimental PL profiles would be deeply affected. Indeed, the WSe₂ related PL intensity while the excitation is taking place would not go to zero as a new source of exciton able to cross the barrier would now exist. As it is not observed in our experimental data, we can say with some confidence that this process is negligible.

To make this point clearer we modified the following sentence in the Supplement:

‘(2) In the low-density regime, bright and dark excitons are non-interacting particles.’

To

‘(2) In the low-density regime, we considered that bright excitons are non-interacting with other type of excitons mainly: dark excitons, B-excitons, momentum forbidden excitons.’

And in the main text line 184:

‘Finally, we considered dark and bright excitons as non-interacting species.’

To

'Finally, we considered that bright excitons are not interacting with other type of excitons (dark excitons, B-excitons, momentum forbidden excitons).'

- In the definition of the partition coefficient, κ , in equation 3 it appears the same exciton mobility and diffusion coefficients are used for both materials. Please provide explanation of why the same coefficients can be used for both materials given differences in electronic and structural properties.

We thank the referee for this comment. It is important to clarify that the mobility (μ_b) and diffusion coefficients (D_b) that appear in the definition of kappa (equation 3) are the exciton mobility and diffusion coefficient inside the interface, and not the mobility and diffusion coefficient of the individual WSe₂ and MoSe₂ layers. (please see line 191 describing the local constant flux density in the text). However, the partition coefficient indeed depends on excitonic parameters of the individual layer through the diffusion length of each material, namely L_1 and L_2 (see equation 3). These parameters are indeed found by the fitting process to be different (see table I). We emphasize that the strength of our model lies in the fact that the values of μ_b and D_b do not need to be known beforehand to determine kappa as it is determined through the fitting process.

- Given the importance of the junction location to the interpretation of the data (e.g. the definition of $X_{\text{tip}} = 0$ in Fig. 2), the authors should provide further description of how the junction location is determined with tip enhanced Raman spectroscopy. It would be illustrative if the authors could provide a supplementary figure to this effect.

We have incorporated an additional figure to the supplementary information. Figure SI 6 provides the method used to determine the position of $X_{\text{tip}} = 0$. The position of the interface is determined by pinpointing the x_{tip} position where the signal abruptly change from MoSe₂ Raman signature to WSe₂ Raman signature.

Minor revisions:

- Perhaps I am misunderstanding the meaning, but it seems that the position-dependent partition functions, κ_+ and κ_- , are dispensed with on line 193-194. What is the reason for defining the discontinuous partition function for different spatial regions in equation 3 if it is then immediately disregarded ("...we will refer simply to κ independently of the source position")?

We thank the referee for the thorough reading of the text. In fact, κ_+ and κ_- are intrinsic constants of the junction depending on the materials forming the junction and the direction of excitonic flux through the junction because of the asymmetrical behavior of the junction. For this reason, κ_+ and κ_- are not mathematically forced to be equal, so to be thorough, in the initial description of the partition coefficient (on line 193-194), we decided to distinguish each term. However, in our specific case, as the partition coefficient we deduced from the fit of the PL is very low, due to a very efficient barrier, the value of κ_+ and κ_- are then indeed nearly equal. To simplify the discussion, all the following reference to the partition coefficient are made to a unique κ independently of the direction of excitonic flux.

We changed the text from: (line 206)

This partition coefficient $\kappa = n(x = 0+)/n(x = 0-)$ can be written, depending on the excitation position, as:

κ quantifies the discontinuity of the exciton density at the junction, with $\kappa_+ = \kappa(x_{\text{tip}} > 0)$ describing the exciton diffusion from WSe₂ to MoSe₂ and $\kappa_- = \kappa(x_{\text{tip}} < 0)$ the diffusion from MoSe₂ to WSe₂. In our case, as the local equilibrium is established at the steady state, $\kappa_+ \approx \kappa_-$, and to simplify the discussion we will refer simply to κ independently of the source position.

To:

"Because of the asymmetrical behavior of the junction, this partition coefficient $\kappa = n(x = 0+)/n(x = 0-)$ can be written, depending on the direction of excitonic flux, as follows:"

"with L_1, L_2 the diffusion lengths of both materials. κ quantifies the discontinuity of the exciton density at the junction, with $\kappa_+ = \kappa(x_{\text{tip}} > 0)$ describing the exciton diffusion from WSe₂ to MoSe₂ and $\kappa_- = \kappa(x_{\text{tip}} < 0)$ the diffusion from MoSe₂ to WSe₂. In our case, as the local equilibrium is established in the steady state, $\kappa_+ \sim \kappa_-$ (see table I), to simplify the discussion we will therefore refer simply to κ independently of the direction of excitonic flux"

- Given the prominence of the analogy to thermal Kapitza resistance in the title, I suggest introducing the analogy in the introduction (perhaps after line 56) rather than on page 10.

To introduce the concept of thermal Kapitza resistance sooner in the text, we modified the introduction from: (line 74)

"We show that the difference in the energy gap at the LH generates a discontinuity in the exciton density distribution"
To

"We show that the difference in the energy gap at the LH generates a discontinuity in the exciton density distribution in analogy to the temperature discontinuity found at interfaces presenting thermal resistance (Kapitza resistance)"

- To better illustrate the exciton diffusion process taking place, I suggest creating a diagram of the different excitonic processes taking place when excitation occurs on either side of the junction.

Thank you for this suggestion to improve our manuscript. A diagram has been added to figure 2 illustrating the different type of excitonic species that are observed in our measurement and their relative position in energy.

- In the discussion of the spatially-averaged PL starting on line 87, the authors describe how the PL spectra of the un-LH differs from the e-LH spectra, including increased peak widths, intensity, and asymmetry. Given the number of differences between the two sample regions, I suggest removing the phrase "nearly identical" on line 88.

Following the suggestion from the referee, the phrase "nearly identical" has now been removed.

- The cause of the increased intensity in the un-LH sample region is initially surprising (line 89) given the lack of the capping hBN and is not discussed in detail until page 12; I suggest adding a statement alluding to this detailed discussion on page 12 in the initial description on page 4.

We thank the referee for pointing out this mistake in the text, in fact the PL described in line 89 is a micro-PL measurement (describing spectra displayed in Figure 1) and not a TEPL measurement (described in Figure 2) and present very similar intensity. We removed the sentence *"with an increased intensity"* from the text.

- An exciting proposal from this study is the ability to control exciton diffusion lengths by deposition of different thicknesses of hBN (or similar materials). Is it possible to replicate the exciton diffusion behavior of the e-LH region in the un-LH by displacing the AFM tip from the surface?

The reviewer is right. It should be possible, as both regions have a very similar L_{eff} vs power density behavior (see figure 4). Carefully controlling the tip-sample distance away from the TMD layer in the un-LH region should result in an excitonic transport with a decreased diffusion length similar to the one observed in encapsulated LH.

- On line 248 and in the discussion of the time-resolved diffusion profile that follows, please include references to the corresponding figures and discussion in the Supplementary text.

We have completed the sentence which is linked to the supplements from *"Experimental conditions and detailed results are given in supplements"* to *"Experimental conditions and detailed results are given in the Supplementary information Fig 2 to 4"*

- The PL spectra in Figure 1 C & F are referred to as "typical far-field PL;" please clarify how this is different than the PL TEPL in Figure 2 and beyond.

Far-field PL spectra are measured using a micro-PL set-up (microscope) (see line 94 for description) while TEPL is measured using the Tip enhanced spectrometer. The main difference is therefore the resolution as micro-PL is limited by the diffraction limit.

To make this point clearer and avoid any confusion we modified the caption of Figure 1 :

"typical far-field PL" by "typical μ -PL spectra"

- Please provide a brief description or reference as to how the AFM tip-enhanced Gaussian excitation profile determined (line 109).

The width of the tip-enhanced Gaussian profile is set to the radius of the silver tip used for measurement, which in turn fixes the resolution of the Tip-enhanced optical measurement. However, the amplitude doesn't need to be fixed because the fitting is conducted on the normalized PL.

- When describing the experimental setup, please also provide the pump laser energy (~ 1.96 eV) for better comparison with the PL data presented in eV in the manuscript figures.

We added 'using a 633 nm (~1.96eV)' to the description of figure 1

The sentence at line 120: "Fig.2 A. displays a schematic of the experimental set-up, showing the linearly polarized laser excitation of 633 nm wavelength focused onto the apex of an atomic force microscope (AFM) silver-coated tip." Have been changed to "Fig.2 A. displays a schematic of the experimental set-up, showing the linearly polarized laser excitation of 633 nm wavelength ($\approx 1.96\text{eV}$) focused onto the apex of an atomic force microscope (AFM) silver-coated tip."

- Please define ϵ and L_D in their first usage on line 171.

The sentence line 186:

"As illustrated in Fig.3 A, we model the junction as an ideally thin interface of fixed width $\epsilon \ll L_D$, where no electron-hole pair recombination can occur."

Have been changed to:

"As illustrated in Fig.3 A, we model the junction as an ideally thin interface of fixed width (ϵ) much smaller than the excitonic diffusion length inside the barrier ($\ll L_D$), where no electron-hole pair recombination can occur."

- Please define L_1 and L_2 after equation 3 (line 191) rather than line 209.

'with L_1, L_2 the diffusion lengths of both materials.' Have now been added after equation 3 at line 208.

- Many of the references show an error where the titles are immediately preceded by "en."

References presenting an error have now been corrected.

- Throughout the manuscript, the supplementary information is referred to in different ways.

We now use the term "**supplementary information**" throughout the manuscript

Reviewer #2 (Remarks to the Author):

The manuscript by Lamsaadi et al. presents an interesting study of exciton transport across a lateral TMD heterojunction. The authors use tip-enhanced photoluminescence spectroscopy to study the propagation of photo-excited excitons and draw conclusions from changes in the luminescence spectrum. To complement these measurements, the authors use a classical drift-diffusion model to numerically analyze the experimental behavior. Using reasonable values for input parameters, the authors achieve good agreement between the numerical and experimental results.

Both the lateral heterostructure and the experimental technique employed in this study provide a new perspective on exciton transport in TMD-based structures. The experiments and simulations support the main claims of the manuscript. Thus I believe this manuscript will be of interest to a broader community of researchers working on exciton transport. Therefore I support the publication of this manuscript.

I have a few minor comments:

The lack of hBN layers on top of the heterostructure is presented as an advantage, because it results in higher exciton densities created by the optical near-field excitation. I must say that I am not very fond of this argumentation, since it suggests that inherently changes when one does not use top hBN. As the authors discuss, however, the density-dependent increase in the exciton diffusion coefficient is merely due to effective changes resulting from non-radiative exciton recombination in high-density regions of the excitation area. To justify the claim that the lack of top-hBN actually enhances exciton propagation, the authors would need to present a quantitative comparison of PL intensities observed in the different regions of their e-LH and un-LH samples. They should show that the PL intensity at the interface is indeed larger when the "effective" diffusion coefficient has increased, which requires a careful study of the excitation power-dependence.

First, we kindly wish to point out that the manuscript does not make any claims regarding the enhancement of exciton propagation due to the absence of top-hBN. The paper only claims that due to specific experimental condition link to tip enhanced spectroscopy, the measurements in e-LH and un-LH are done with very different excitation power density (much higher in the case of the un-LH sample). We further show in Figure 6, that in both

e-LH and un-LH the effective diffusion length is increasing with excitation power density, thus explaining the difference observed in TEPL. As it can be seen in Figure 4 at the same power density the effective diffusion length measured for un-LH and e-LH are very similar, showing no superiority of non-encapsulated sample. When writing the manuscript, it was not our intention to present non-encapsulation as an advantage: as shown in figure 1, the optical transitions are broader suggesting a reduced optical quality combined with less protection from unwanted doping. Following the described variation in equation 7, the non-encapsulated sample is just the ultimate limit of electric field enhancement and serve as a case demonstrator. Ideally, it would be interesting to study configurations with 1, 2, more layers of top-h-BN to investigate the near field enhancement effect with a fully protected TMD layer, which is beyond the objective of the present paper.

To make this point clearer we modified the sentence line 288 from:

'As a matter of fact, controlling the thickness of the top hBN layer from a single layer to 20 layers, for example, would change the enhancement factor by a factor 10'

To:

'As a matter of fact, controlling the thickness of the top hBN layer from a single layer to 20 layers, for example, would change the enhancement factor by a factor 10, modifying the diffusion length by the same amount, all the while avoiding the flaws, lower optical quality and exposition to environment, that are observed in the uncapped LH.'

The authors argue that the slow-down of excitons at the interface leads to local increases in the exciton density and the non-radiative lifetime, which explains the observed increase in PL intensity. However, one could argue that non-radiative Auger recombination also increases due to the local enhancement of the exciton density. Maybe the authors can include this point in their discussion.

We thank the referee for this pertinent comment, the text has been modified to include this possibility:

"Other phenomenon may contribute to this variation as a modification of the non-radiative Auger recombination due to the local enhancement of the exciton density"

Reviewer #3 (Remarks to the Author):

In this manuscript, the authors described a Kapitza-like-Resistabnceexction dynamics in a 2D heterogeneous structure. The results reported an excellent data on the unidirectionalexctonic transport, showing a huge progress in this field. In addition, a theoretical model is shown in this work to approval this experimental results. After reviewing, in my opinion, it could be considered acceptable in Nature Communications. Some suggestions are provided for authors to consider:

We thank the referee, for

1. In the introsuction part, the authors should state how to control the exction in BN?

The direct bandgap of bulk hBN (more than a few layer) is 5.95 eV (208.3 nm) which is far above the laser excitation energy, therefore the optical contribution (from excitons for example) from the encapsulating h-BN can be neglected. Since it does not contribute the phenomenon described in the paper and in the interest of brevity, we decided not to tackle this subject.

2. In Fig. 1a, the authors have to distinguish the heterogeneous region of the two materials.

With an optical microscope, as can be seen in Figure 1 A there is no observable difference between the MoSe₂ and the WSe₂ layers, it would therefore be complicated for us to place accurately the separation between the two materials, and we would rather not add inaccurate information to the text. The only way to distinguish the two materials and place the barrier without making error is to use the PL or Raman integrated maps shown In Figure 1 D and G and Figure 5 in Supplement.

3. The color distribution in Figure 1C&G is not uniform, and the intensity varies too much, how to prove that the test area, which truly reflects the material properties?

This is a very pertinent comment thank you: The strong variation of PL intensity come from Bi-layer inclusions as the gap become very slightly indirect. But indeed, other smaller variations of the PL intensity can appear close to

defects, mainly bubbles trapped between the TMD layer and the bottom BN, but other defects as well like ripples or small cracks. The Raman characterization shown in Figure 5 of the SI confirm that the sample is quite homogeneous. The optical far field (Raman and PL) characterization shown in Figure 1 and Figure 5 of the SI were done to determine the area of the samples the more "intrinsic" (with the more homogeneous: PL (both amplitude and energy), Raman (both amplitude and energy) and AFM profile.). Consequent efforts have been dedicated to this and as can be seen in Figure 2 C, E and Figure 6 in supplement, one can see that the variations of PL (both amplitude and energy) and Raman (both amplitude and energy) far from the junction are very small.

4. In line 78, dashed blue should be dashed white?

You are perfectly right, the sentence line 92:

"The dashed blue and yellow line in Fig. 1 A highlight the boundaries of the bottom and top hBN flakes, respectively."

Has been changed to:

"The dashed white and yellow line in Fig. 1 A highlight the boundaries of the bottom and top hBN flakes, respectively."

5. In addition, the authors should add some discussion on the PL spectra, in order to show clearer to readers.

The discussion concerning the PL spectra has been expended in the text, with: The addition of a diagram to figure 2 illustrating the different type of excitonic species and process that are observed in our measurement and their relative position in energy. The text has been modified to comment on this new figure (line 136): *"All TEPL spectra can be fitted using three Lorentzian peaks to account for the previously described relative contributions of $A^{W_{Se_2 1s}}$ (dashed red line), $X^{D_{WSe_2}}$ (dashed green line) and $A^{MoSe_2 1s}$ (dashed blue line) as illustrated by the diagram in Fig 2 A."*

Reviewers' Comments:

Reviewer #1:

Remarks to the Author:

The authors have largely addressed the concerns raised by my previous review, including an expanded introduction to the study of exciton diffusion, clarifications, and additional references. The authors' revisions improve the quality and messaging of the manuscript, and I support the publication of the manuscript as is.

One minor revision that I believe would further enhance the manuscript would be to include a statement comparing the timescales of exciton diffusion in the MoSe₂ layer and across the WSe₂-MoSe₂ junction with the relaxation time of an excited B exciton to an A exciton in the MoSe₂ layer (in the context of my previous question about MoSe₂ B exciton excitation). Whether this statement is included or not, I do not feel further review is necessary.

Reviewer #2:

Remarks to the Author:

All queries have been addressed by the authors in the response letter and the revised version of the manuscript. I fully support publication of this manuscript.

Reviewer #3:

Remarks to the Author:

I recommend its acceptance.

REVIEWERS' COMMENTS

Reviewer #1 (Remarks to the Author):

The authors have largely addressed the concerns raised by my previous review, including an expanded introduction to the study of exciton diffusion, clarifications, and additional references. The authors' revisions improve the quality and messaging of the manuscript, and I support the publication of the manuscript as is.

One minor revision that I believe would further enhance the manuscript would be to include a statement comparing the timescales of exciton diffusion in the MoSe₂ layer and across the WSe₂-MoSe₂ junction with the relaxation time of an excited B exciton to an A exciton in the MoSe₂ layer (in the context of my previous question about MoSe₂ B exciton excitation). Whether this statement is included or not, I do not feel further review is necessary.

We thanks the referee, it is indeed a good idea. Unfortunately acquiring the necessary knowledge, to compare the timescales exciton B and A diffusion across the WSe₂-MoSe₂ junction would involve a whole new set of experiment campaign focused on the relative lifetime and intensity of both excitonic species. This will be one of the focuses of our next studies.

Reviewer #2 (Remarks to the Author):

All queries have been addressed by the authors in the response letter and the revised version of the manuscript. I fully support publication of this manuscript.

Reviewer #3 (Remarks to the Author):

I recommend its acceptance.